# Control Theory and Systems Biology: Potential Applications in Neurodegeneration and Search for Therapeutic Targets

**DOI:** 10.3390/ijms25010365

**Published:** 2023-12-27

**Authors:** Andrea Angarita-Rodríguez, Yeimy González-Giraldo, Juan J. Rubio-Mesa, Andrés Felipe Aristizábal, Andrés Pinzón, Janneth González

**Affiliations:** 1Departamento de Nutrición y Bioquímica, Facultad de Ciencias, Pontificia Universidad Javeriana, Edf. Carlos Ortiz, Oficina 107, Cra. 7 40-62, Bogotá 110231, Colombia; mangaritar@unal.edu.co (A.A.-R.); yeimy.gonzalez@javeriana.edu.co (Y.G.-G.); andres_aristizabal@javeriana.edu.co (A.F.A.); 2Laboratorio de Bioinformática y Biología de Sistemas, Universidad Nacional de Colombia, Bogotá 111321, Colombia; ampinzonv@unal.edu.co; 3Departamento de Estadística, Facultad de Ciencias, Universidad Nacional de Colombia, Bogotá 111321, Colombia; jjrubiom@unal.edu.co

**Keywords:** control theory, systems biology, neurodegenerative diseases, genome-scale metabolic networks

## Abstract

Control theory, a well-established discipline in engineering and mathematics, has found novel applications in systems biology. This interdisciplinary approach leverages the principles of feedback control and regulation to gain insights into the complex dynamics of cellular and molecular networks underlying chronic diseases, including neurodegeneration. By modeling and analyzing these intricate systems, control theory provides a framework to understand the pathophysiology and identify potential therapeutic targets. Therefore, this review examines the most widely used control methods in conjunction with genomic-scale metabolic models in the steady state of the multi-omics type. According to our research, this approach involves integrating experimental data, mathematical modeling, and computational analyses to simulate and control complex biological systems. In this review, we find that the most significant application of this methodology is associated with cancer, leaving a lack of knowledge in neurodegenerative models. However, this methodology, mainly associated with the Minimal Dominant Set (MDS), has provided a starting point for identifying therapeutic targets for drug development and personalized treatment strategies, paving the way for more effective therapies.

## 1. Introduction

Over the years, the growing knowledge on the key molecular and cellular mechanisms of many diseases has led to the transformation of medicine into a proactive discipline that is focused on being predictive, personalized, preventive, and participatory, which is known as P4 medicine. This approach aims to achieve the early diagnosis of human diseases and to reduce costs in the health sectors [1]. However, for most pathologies, the currently used therapies have been developed based on experimental trials somehow limited by a traditional reductionist approach and clinical trials involving large cohorts, where it is usually assumed that participants will all respond similarly to the same environmental or molecular perturbation. In most cases, this approach does not take into consideration that, for instance, genetic diversity and environmental factors can trigger differential responses among individuals [2]; this situation could cause difficulties in the successful identification of treatments, which also depends on the complexity of the disease itself. Therefore, there is an increasing need for the identification of reliable targets that will enhance our predictive power [3,4].

Complex diseases such as neurodegenerative diseases (NDs) lack therapies to stop their progression, as they involve multiple pathogenic determinants, where genetic, environmental, and behavioral factors contribute to the progressive cellular death and loss of neurons in the brain [5,6]. This is of course the case for the major NDs studied so far, including Alzheimer’s disease (AD), amyotrophic lateral sclerosis (ALS), Parkinson’s disease (PD), and Huntington’s disease (HD) [7,8,9]. In general, traditional studies have determined that these diseases share some characteristics [6]. Therefore, experimental analyses and studies have been focused mainly on some of those few well-known key elements; this type of approach has been derived from our partial understanding of the mechanisms and pathways associated with these types of pathologies [10]. The above means that therapies currently intend to reduce or lessen the severity of ND diseases; however, they do not target the causal elements of the pathologies but rather those merely related to, for instance, their symptoms. However, an optimistic future for revealing causative pathogenic events relies nowadays on the integration of highly multidimensional omics (i.e., genomic, transcriptomic, and proteomic data) coupled with modern computational approaches, such as systems biology approaches, capable of handling large biological datasets and finding subtle networked relationships between heterogeneous information [6].

Neurodegenerative diseases present a high heterogeneity, which means that there is a high burden in their progression, pathology, and clinical presentation [11,12,13]. The heterogeneity observed in neurodegenerative diseases is due to their multifactorial causes, such as genetic factors, lifestyle, and even environmental factors [13]. Multiple studies have determined that this heterogeneity is a confounding factor for understanding these diseases, and, consequently, the development of treatment has been a great challenge to date [11,12,13,14].

In this sense, due to the complexity of NDs and their heterogeneity, the advent of omics, the increasing availability of omics data, and the widespread plethora of computational tools in databases have allowed systemic studies to better understand the mechanisms associated with the biological response to diseases [15,16,17]; this has enabled the development of a comprehensive vision of complex phenotypes, exemplified by computational representations of whole cellular metabolism, such as in the case of genome-scale metabolic networks (GEMs) [18,19,20,21]. These GEMs, which are mathematical representations describing a complex set of mass stoichiometrically balanced metabolic reactions of an organism or cell, have been established as one of the main modeling approaches for systemic studies [22,23]. In addition, this approach also permits the use of annotation associations that serve as the basis for the integration of omics data [24] and make use of linear optimization techniques to integrate models and therefore predict cellular behaviors, as well as underlying biochemical mechanisms of disease [25].

There are different approaches to perform GEMs. Conventional methods used to integrate them have shown to provide a biased view of a phenotype, since they restrict the solution space to a reference state, which limits the impartial understanding of the dynamics and regulatory mechanisms of biological systems [26,27]. Some authors have illustrated the selection bias of COBRA methods, including methods such as flow balance analysis (FBA), which require the selection and optimization of a function to identify the metabolic phenotype [28,29]. FBA is the most basic biased method used for simulating metabolism [30], since it optimizes via linear programming, finding a solution in the solution space of a single possible metabolic phenotype [21]. However, in some cases, the metabolic phenotype identified may not reflect the reality of the metabolic network [28,30].

To overcome this limitation, a methodology is required that can take into consideration the state variables of a high-dimensional system, such as a complex biological network. Furthermore, it is important that such a methodology can identify the groups of minimal controlling reactions that directly or indirectly control the cell system behavior without prior knowledge of cell targets. This goal can be achieved through the application of control theory [31,32,33,34,35]. In this context, control theory methods provide an ideal framework for analyzing the ability of complex networks to intervene in the biological system through appropriate control signals [35]. However, few studies have used these control methods on multi-omics GEMs to predict controller reactions and their possible implications as therapeutic targets for the diagnosis, prevention, and treatment of chronic diseases [26,36].

Considering the above, this review examines the most widely used control methods in conjunction with genomic-scale metabolic models in the steady state of the multi-omics type. In addition, this work provides an overview of their applications in human chronic diseases, and its possible extrapolation in processes of cognitive deterioration is proposed.

## 2. Results and Discussion

### 2.1. Systems Biology, Genome-Scale Metabolic Models (GEMs), and Omics

Systems biology aims to describe and analyze the emergent properties that arise from complex interactions in a biological system [37]. From an interdisciplinary perspective, systems biology has been driven by computational advances that use mathematical models to perform quantitative molecular measurements and predictions [20,38]. Although systems biology is still considered a recent approach for the study of biological complexity, it has provided a powerful means for obtaining a holistic understanding of biological systems, including cells, organisms, and even communities [39,40]. This has led to a shift in the scientific approach in which the contextualization of computational modeling with omics data (transcriptome, proteome, and genome data, among others) offers a better understanding of cell growth, adaptation, development, and even disease progression [41].

The advent of omics through technological advances such as transcriptome sequencing (RNA-seq) [42], SNP-chip profiling [43], and whole-genome or meta-genome next-generation sequencing [44,45], among others, has largely ensured a more approximate description of biological reality [38]. Nevertheless, given the large amount of data typically generated by these methodologies, as well as their heterogenous nature, there is a lack of approaches that allow for the integration of this information into a complete and holistic view of the biological system under study, which remains a significant challenge that represents a bottleneck for all multi-omics studies.

To date, perhaps one of the most reliable approaches being widely implemented for such data integration into a holistic view of the metabolic phenotype is GEMs [20]. GEMs are mathematical representations of cellular metabolism that not only encompass our understanding of the physicochemical states of cells but also have the capability to incorporate dynamic/kinetic information. Depending on the specific case, these models can be solved using systems of equations (differential or partial) [31,40,46], or they can be steady-state models that are based on the principle of mass balance and do not require kinetics information [47,48]. Regardless of the type of modeling, the biological system is represented as a set of metabolic reactions and extensive data related to associated cellular behavior rules that facilitate predictions of genotype–phenotype relationships (Figure 1) [40,49,50].

The most well-known association rules in GEMs are the gene–protein–reaction (GPR) rules that, in general, describe the set of gene expression dependencies necessary for a biochemical reaction to take place inside a cell. In these GPR rules, each reaction is associated with one or more enzymes that are, in turn, encoded by a specific gene or a set of genes, depending on whether the reaction-catalyzing enzyme is coded for by one, two, or more genes. Overall, these sets of rules are the bases for the integration of different omics data into GEMs [51,52], since it is possible to translate transcriptomic, proteomic, or even metabolomic data into different gene state rules. This approach has been widely used in the clinical field, where studies have focused on identifying biomarkers and predictive targets, as seen in the diagnosis of congenital errors [53], neuropsychiatric diseases [54], or markers of hypoxia in erythrocytes [55,56], among others. However, there are still many challenges for system modeling and omics integration [40], mainly because this integration into GEMs (especially dynamic ones) is typically limited to well-defined systems.

Therefore, the use of steady-state models has shown to be a good approximation strategy to estimate the study of the metabolic phenotypes of cells or organisms according to their environmental and intrinsic conditions [57,58], and, hence, these models can be integrated by the same means. This integration is generally performed by means of a linear optimization approach [59]. In this regard, one of the most widely used methods to integrate these models is FBA (Figure 2A) [60,61]; this method assesses intracellular fluxes to predict cellular metabolic changes under thermodynamically feasible states, considering specific environmental conditions, which help constrain the optimization solution space [4,59,62,63], and, thus, it better reflects the biological conditions of the system. Overall, these conditions aim to optimize the biologically relevant objective function for the cellular metabolism under study [40].

While GEMs have traditionally been restricted to the study of metabolic changes for single-cell simulations, at least from 2013 [64], several research efforts have been made to extend this approach not only to the study of whole cellular communities but also to the incorporation of several layers of biological information into a single modeling framework. This has inspired the use of a wide variety of complementary analyses, including functional enrichment [65], protein–protein interaction [66], and gene regulatory network analyses [67], among others. The systems biology approach has played a pivotal role in laying the groundwork for the development of multilayer networks, which have led to the prediction of, for example, dietary supplements for Crohn’s disease with metagenomic data [68], the modeling of the effect of drugs available on the market in human metabolism and their multitarget effects on diseases of interest [4], and studies of the human intestinal microbiome and its relationship with specific effects on multiple organs and chronic diseases [69].

The preceding provides an insight into the usefulness of the systems biology approach and GEMs to (1) integrate regulatory information, as well as other types of biological information, into a single computational framework and (2) transform this information into regulatory and biological rules that better reflect multiple cellular processes at different time and spatial scales.

These characteristics of GEMs have allowed them to become integration tools for constantly improving omics data and cellular behavior prediction, expanding not only the field of bacterial biotechnology applications but also the field of personalized and precision medicine in more complex organisms, such as humans.

### 2.2. Key Steps for Performing a Reconstruction of Genomic-Scale Metabolic Models (GEMs)

The reconstruction of genome-scale metabolic models (GEMs) is a systematic process involving several key steps. The process is divided into (a) genome annotation, (b) environment specification, (c) biomass formulation, and (d) model curation (the elimination of leakage metabolites and the checking of gaps, among others) [25,70]. GEMs provide a framework for mapping species-specific knowledge and complex omics data to metabolic networks, and when combined with constraint-based reconstruction and analysis (COBRA) methods, such as flux balance analysis (FBA), they facilitate the translation of hypotheses into algorithms that can be used to generate testable predictions of metabolic phenotypes [21,25]. These algorithms allow for the generation of testable predictions of metabolic phenotypes, such as growth rates or responses to environmental conditions. The ability to experimentally validate these predictions reinforces the utility of GEMs in biological research, providing powerful tools to explore systems biology, formulate new hypotheses, and advance practical applications in biotechnology and medicine [4,20].

#### 2.2.1. Inference of Enzymes, Reaction Directionality, and Compartments

As mentioned above, the first stage in building a genome-scale model involves annotating the genome to generate an initial list of functions or reactions. In this process, open reading frame regions are searched for and identified in the genome of the sequenced organism, which is linked to the relevant gene–protein–reaction rules (GPRs). These GPRs are associated with specific enzymatic behaviors in a substrate, which can generate key metabolites [21,63].

According to Pitkanen and collaborators [71], an essential part of the specification of a metabolic model intended for computational analysis involves the determination of the reversible and irreversible transport reactions, the corresponding compartments, and connectivity, that is, the metabolic pathways that the model must contain. Two approaches related to enzyme prediction and compartmentalization stand out: automatic prediction and transfer annotation, which are derived from the reconstruction approach using comparative or exploratory methods [20,71].

The comparative method is used to predict metabolic pathways based on a previously functionally annotated genome or an existing model of a standard organism. This technique uses enzyme hierarchies (ECs), gene names, and generated products as input, followed by preliminary curation, which involves comparing information in databases from various species [72,73].

The selection of reference pathways is driven by the Bayesian probability of occurrence. This decision considers the pathways’ availability in databases and their suitability for the model. Despite its merits, this method can introduce information gaps stemming from the absence of reactions or enzymes unique to the target model. To address this, the Bayesian probability model incorporates additional information, such as BLAST scores, genomic context, and functional data [74,75].

Functioning as an annotation-by-reference framework, this approach involves annotating and curating the target genome in comparison to a reference genome. Furthermore, an alternative methodology, rooted in the comparative method, has been proposed. This method leverages orthologous associations between the genome of the standard organism and that of the target organism. This innovative approach enables the inference of a new model for a closely related or phylogenetically linked species. It capitalizes on the correspondence between organisms to establish connections between metabolites across compartments through gene associations and reversible enzymatic reactions, as outlined in prior studies [76,77].

Therefore, it is important to define the biomass objective function. The biomass objective function is a mathematical representation of the growth requirements of a cell that describes the rate at which all of the biomass precursors are made in the correct proportions. It is used to computationally predict cell growth [20].

The growth rate can be obtained from the biomass flux by using the biomass yield coefficient (*Yx*/*s*), which is the ratio of the biomass produced to the substrate consumed. The growth rate (*μ*) is equal to the biomass flux (*v*) divided by the biomass yield coefficient:(1)$$Y\frac{x}{s},\;or\;\mu =v/Yx/s$$

Therefore, predicting the flux rate of organisms involves the simulation of metabolic fluxes in the system, an essential process that directly influences growth rate predictions [78]. Predicting the flux rate of organisms involves sophisticated methods, such as metagenomic growth estimators, which, together with genome-scale models, represent essential approaches for understanding microbial dynamics in diverse ecosystems [78,79]. Metagenomic growth estimators have been designed to estimate the growth rates of microorganisms using the intrinsic characteristics of microbial genomes and discrete metagenomic samples, allowing for non-invasive assessments of microbial expansion in natural environments [80,81].

The estimation of specific growth rates involves kinetic models based on cellular growth kinetics, enabling the prediction of growth rates across various organisms and conditions. Gene expression signatures enhance prediction accuracy by assessing the expression of a specific gene set [82,83]. Optimal growth temperature prediction utilizes genomic sequences to predict growth temperatures in prokaryotes, leveraging genomic adaptations to their environment [30,84]. Additionally, machine learning techniques, such as k-nearest neighbors’ regression, facilitate the construction of predictive models for growth rates using gene expression data [83]. These diverse methods, ranging from genomic data-driven approaches to machine learning approaches, showcase the precision and versatility of contemporary strategies for predicting flux rates in diverse biological contexts.

#### 2.2.2. Tools for Genomic-Scale Metabolic Model Reconstruction

In the process of reconstructing genome-scale metabolic models (GEMs), both automatic and semi-automatic tools are used. These tools play a crucial role in automating and facilitating the compilation of information necessary to build detailed models of the metabolic networks of organisms [25,63,85]. Automatic tools perform reconstruction tasks fully automatically, while semi-automatic tools provide an interactive interface that allows user intervention in specific steps of the process. This combination of automated and semi-automated approaches streamlines GEM reconstruction, enabling more efficient and accurate analyses of complex metabolic interactions at the genomic level [86,87].

The tools employed for reconstruction encompass a variety of widely used methods, as illustrated in Figure 3.

RAVEN Toolbox (Version 2.8.6.0) (Reconstruction, Analysis, and Visualization of Metabolic Networks): RAVEN, a MATLAB-based tool, is critical in enabling constraint-based metabolic modeling. It facilitates the semi-automatic reconstruction of preliminary de novo models for specific organisms from the genomic sequence. RAVEN offers two distinct approaches to initiate GEM reconstruction: based on protein homology to an existing template model or de novo using reaction databases [88].GPRuler (Version 3.7): An open-source framework, GPRuler (Version 3.7) efficiently automates any living organism’s GPR rule reconstruction process. This framework has been validated in various case studies, demonstrating its ability to reproduce original GPR rules with high accuracy. The applicability and accuracy of GPRuler make it a valuable tool for generating accurate models [89].Methods for the automated reconstruction of genome-scale metabolic models: These methods, addressed in a review, discusses various tools and algorithms for the rapid reconstruction and analysis of metabolic models at the genomic scale. They highlight the importance of GEM reconstruction in supporting predictive analysis and the characterization of genomes based on sequence data. These reviewed and critical approaches provide a comprehensive view of the current state of available tools [90].

The reconstruction of genomic-scale metabolic models (GEMs) utilizes automatic and semi-automatic tools, with the COBRA Toolbox (Version 2.8.6.0) being a prominent choice. This MATLAB-based toolbox, representing constraint-based analysis and reconstruction, offers specific functions for modeling and simulating biological systems. Functions like flux balance analysis (FBA) and other constraint-based methods are available within the COBRA Toolbox, empowering researchers to efficiently reconstruct and analyze metabolic networks [25]. These streamlined tools enhance the reconstruction process, providing researchers with accessible and powerful resources for understanding genomic-scale metabolic systems.

### 2.3. Classification of Genome-Scale Models: Steady-State and Dynamic Models

Throughout this review, we highlight significant advances in understanding the interactions between molecular components in an organism. Currently, it is possible to infer, to some extent, the topology of a metabolic network, although the parameter dependence and dynamics of such networks still generate ongoing debates [78]. Nevertheless, the connection of observations has been achievable, such as the flux rate, control mechanisms, and heterogeneity in genetic expression, through a mathematical description. This goes beyond optimal metabolic fluxes, addressing the joint distribution of metabolic fluxes [78]. Consequently, the modeling of metabolic networks can be approached in two ways depending on the amount of experimental information available: through static or dynamic models.

#### 2.3.1. Steady-State Metabolic Models

Although the study of static models and their response to stimuli presents challenges, this approach helps simulate the perturbation of a system in a stable or homeostatic state [91,92]. Formulating static models in a system can be achieved in two ways. In the first, enzymatic reaction rates equal to zero are assumed, which describe, for example, thermodynamic cycles in equilibrium. According to their functional annotation, this approach is known as the null space to the right of the stoichiometric matrix containing the flux distributions of enzymatic reactions [78].

As detailed later, the stoichiometric matrix, *S*, represents a linear transformation of a flux vector to a time vector, *t*, derived from a concentration vector that is equal to zero, thus defining the null space [21,93]:(2)Sv=0

Then, in the steady state, all flow distributions reside in the null space, defined with a dimension of *n*-*r*. In this model, there is no accumulation or depletion of mass, meaning that the production rate is equal to the consumption rate in the metabolic network. This balance is expressed by Equation (2) [22,93]. The numerical integration of the mass balance (stoichiometry) allows for the specificity of the stimulus to be analytically analyzed according to the study’s objectives.

Mass balance models are based on identifying a subset of elements and evaluating them as variables that can be expressed as linear equations [91,92]. These equations span an infinite number of bases for a linear space, and methodologies such as Singular Value Decomposition (SVD), which provides unbiased information about all subspecies of the stoichiometric matrix (*S*), can be used (Figure 4).

Static network modeling aims to predict interactions between drug molecules and target proteins through shared components, facilitating information transmission across network layers. For instance, diseases can be linked via shared genetic associations, gene-disease interactions, and disease mechanisms. This approach establishes connections between diseases through shared genes, enabling drug repurposing opportunities [94]. In the study in [95], a static network model predicted the phenotypic effects of perturbations in biological networks, focusing on gene, protein, and drug interactions. The model utilized three networks: EGFR/MAPK and PI3K/AKT from an experimental study, the TNF regulatory network from the STRING database, and a comprehensive NCI-selected pathway network from the interaction database proteins. The algorithm, based solely on the static network structure, predicted the regulatory effects on proteins/genes when perturbed (e.g., by inhibiting a drug). Despite its simplicity and lack of temporal dynamics, the algorithm demonstrated surprising effectiveness, accurately predicting protein/gene upregulation or downregulation in up to 82% of cases [95].

However, a potential limitation of static network modeling is that dynamic metabolic behaviors in patients may lead to changes in gene expression levels, which may invalidate interactomics in static modeling [20]. This limitation underscores the need for dynamic models to map regulatory relationships between molecules, especially in the context of changes in gene expression levels.

#### 2.3.2. Dynamic Models: A Comprehensive View

For the most part, the set of reactions in a metabolic model is described by differential equations, allowing for the dynamics of the model processes to be inferred. Biological networks, rich in feedback, facilitate the development of dynamic models (Equation (2)). These models are based on reaction rate equations, which describe the temporal change in molecular populations in the system [23,78]. The rate equations follow the rule that reactions that affect the concentration of a species are reflected as source terms. In [96], the equivalence of deterministic and stochastic methods and the importance of stochastic fluctuations were demonstrated, especially in small systems.
(3)$$\frac{dx}{dt}=S\times v$$

Computational advances like the Monte Carlo dynamic probability method and tau-leaping algorithms have improved modeling and analysis. These models allow for the construction of multiscale representations of biological networks, integrating knowledge of specific static and dynamic component functions [97]. Dynamic models are helpful in various protein analyses, such as understanding crop growth and considering plant carbon accumulation and carbon source utilization models. However, we will not go into them in detail.

Applying algorithms and dynamic models in biology provides dynamic information, such as gene expression in different pathways. It extends to fields such as the study of crop growth, highlighting their essential role in the understanding of complex biological processes.

### 2.4. Topological Parameters for Identifying Drug Targets in an Enzyme-Centric Network

In systems biology, the study of molecular interactions often involves the analysis of topological parameters such as motifs and groups within biological networks. Motifs, recurring patterns in these networks, and clusters, groups of densely connected nodes, offer insights into functional relationships [98,99].

Analyzing topological parameters, such as centrality, motifs, and clusters, and identifying drug targets in enzyme-centric networks have been revealed as crucial tools for anticipating targets in biological systems. These parameters include global properties that identify significant nodes, such as centers and clusters, and groups of nodes that appear more frequently (motifs) and are more closely connected (clusters). Representing drug–target interactions in a network improves the understanding of complex relationships, and applied topological methods leverage theories of brain self-organization [98]. Studying perturbation patterns in biochemical networks provides opportunities for drug development, and combining algorithms using ensemble approaches improves prediction. Integrating multiple topological methods is a promising strategy to advance drug target prediction, underscoring the importance of holistic approaches in future research [99,100].

In a study on cancer metabolic networks, drug targets were identified using an enzyme-focused network clustering analysis. The results showed that the drug targets gathered in a specific group of an enzyme-centered network of cancer cells. However, authors such as [99,101] recently used topological methods to predict drug targets by taking advantage of a brain network self-organization theory. The authors demonstrated the application of topological methods to predict drug targets by leveraging a brain network self-organization theory, highlighting a network representation of drug–target interactions in a biosystem to improve the understanding of the multifaceted modes of action of drugs and to suggest a therapeutic change for approved drugs. The results showed that the topology adequately exploited by the local community paradigm (LCP) theory, initially detected in the topological self-organization of the brain network and generalized to any complex network, can suggest highly reliable predictions, comparable to the state-of-the-art supervised methods [101].

The above underlines the potential of topological methods, particularly LCP theory, to predict drug targets within biological networks and demonstrates the relevance of such approaches in systems biology and drug discovery. These methods can reduce reliance on labor-intensive experimental approaches and provide a more complete understanding of drug mechanisms of action. However, limited search results specifically address the combination of topological parameters, enzyme-centric networks, drug targets, and the brain.

### 2.5. Genome-Scale Metabolic Models (GEMs) and System Controllability

Control theory is a multidisciplinary framework that combines engineering and mathematics to describe the behavior of dynamic systems [102,103]. In the mathematical branch, control theory provides a basis for understanding how systems respond to inputs and how to manipulate those inputs to achieve specific outcomes [104,105]. In biology, living organisms are complex systems that exhibit intricate control mechanisms to carry out physiological functions, respond to stimuli, and maintain homeostasis. Therefore, this theory offers a way of analyzing a metabolic system in terms of metabolic pathway signaling and genetic interactions—either from the internal state of the system or under environmental perturbations [106].

In the 1970s, research groups led by Michael Savageau and Henrik Kacser elucidated methods for analyzing the control of metabolic fluxes and enzyme activities using a local parametric sensitivity analysis [106]. However, it was the group of Henrik and Burns that applied a standard linearization technique to address the field of steady-state models, which is called metabolic control analysis (MCA) or metabolic control theory (MCT) [106,107,108].

In 2009, ref. [109] established a connection between metabolic control analysis and control theory. In this pioneering work, the researchers explained how the overall theory aimed to link steady-state changes in individual pathway components to steady-state changes in the systematic behavior of the network through a methodology called metabolic control analysis (MCA). The theory was motivated by the rationalization that metabolic flux is not controlled by one rate-limiting enzyme, but rather the control is shared by all, or perhaps a significant subset, of the enzymes in the network. MCA quantitatively rationalizes the underlying mechanisms by which an enzyme exerts a degree of control over the concentration of a metabolite shared between metabolic pathways [107]. This is determined by flow and concentration control coefficients [106,110,111]. Therefore, we can define MCA as a tool whose central concept is the notion of control coefficients (coefficients that quantify the sensitivity of a particular variable, for example, the concentration of a metabolite or a flow rate). Since then, MCA has become a powerful tool in systems biology, applied to understand various biological processes, including cellular metabolism.

### 2.6. MCA Can Be Applied to Both Dynamic and Steady-State Genome-Scale Metabolic Models

Numerous investigations in metabolic engineering have focused their efforts on understanding dynamic control, guiding the options for the possible functioning of various metabolic systems [109,112,113]. In general, dynamic metabolic control offers advantages over other control strategies, which is why it has been a widely used approach [109]. There are multiple perspectives of dynamic control, such as open-loop control, which involves predetermined metabolic control actions that are not necessarily feedback-based [114]. Moreover, authors such as [113] have established a two-step methodology involving multiple stages of regulation. This methodology determines homeostatic cell patterns induced (under conditions of interest) to activate production pathways that would otherwise retard growth [115].

Regardless of the type of dynamic control approach, multiple authors have noted that dynamic control offers advantages over static control by allowing real-time adjustments and the optimization of metabolic fluxes in order for systems to adapt to variable conditions, to optimize productivity more effectively, and to improve the robustness and stability of metabolic pathways [109,115]. Furthermore, the advantages of using dynamic models are associated with the depth perspective of cellular processes and their responses to disturbances [116]. Likewise, the combination and integration of omics data help to unravel cellular processes and to obtain detailed information of the process, enabling computational simulation experiments to be carried out and addressing biological questions in a controlled and reproducible manner [112].

In 2021, ref. [117] described in detail that dynamic metabolic control can be applied to a diverse set of metabolic pathways that regulate metabolite concentrations over time [109,118]. This involves conservation relationships and external parameters, among others, leading to the characterization of control and connectivity matrices that can be used to model the enzymatic kinetic uncertainty of biological models [119].

Examples of biological processes that have been better understood through dynamic modeling include metabolic pathways and regulation, providing insights into the dynamics of metabolic fluxes and their coordination [116,120]; transitions between different phases of the cell cycle [112]; the dynamics of the immune response (including interactions between immune cells, cytokine signaling, and the regulation of immune system activation) [118]; and brain electrical activity models, where information has been obtained through the complex patterns of neuronal activity [103,121,122].

Dynamic metabolic models have their own limitations when applying control theory. First, the computational complexity, particularly given by ordinary differential equations (ODE), demands substantial time and computing power [106,123,124]. Second, parameter estimation requires well-curated estimates of kinetic parameters, which can be challenging due to the lack of available information to date, introducing additional uncertainty [106].

However, recently, researchers have opted to use steady-state metabolic models. The advantages of working with steady-state models and control theory lie in their “simplicity” since it is assumed that the system has established an equilibrium, simplifying the mathematical representation and reducing computational power [125]; computational efficiency implies a reduction in time and computational resources compared to dynamic models. Additionally, steady-state models can predict flux distributions in the metabolic network under steady-state conditions with high statistical confidence [123].

Steady-state models at the genome scale can be analyzed using control theory. MCA provides tools and techniques for understanding biological control strategies to regulate metabolic fluxes and optimize system performance [105,124]. In recent decades, methods have been developed to determine the control elements of a system in a stationary state. Recently, authors such as [29,32] have used approaches that involve flow coupling between reactions, where the activity of one reaction is controlled by directly manipulating another coupled reaction.

Constraint-based modeling provides a framework to investigate metabolic states and define metabolic phenotypes through multi-omics data integration; it imposes known biological constraints to limit the solution space. This provides a powerful tool that allows for inferences about biological reality and makes it a good strategy for understanding the control mechanisms of biological systems. However, few studies have implemented control theory with steady-state models. Therefore, in the next sections, we show the elements associated with control theory, its methodologies, and its applications, emphasizing genomic-scale models in the steady state.

### 2.7. Control Theory Elements and Classification of Nodes in Complex Networks

In the context of genome-scale metabolic models and control theory, several elements and types of nodes are relevant. For instance, nodes can be classified into two broad categories based on their properties and functions in the metabolic model [32,126,127]. As we discuss later, these categories are determined through methodology and data analysis. First, “driven nodes” are those that are influenced or controlled by other nodes but do not directly control any other node, and, second, “controller nodes” are sites that exert control over other nodes [128]. Control sites are reactions whose contribution is quantified in the model of interest in the distribution of metabolic fluxes (Figure 2B) [129]. Authors such as [33,129] have noted that the manipulation of control sites may potentially have a distributed action in several reactions that will lead to a potential effect on metabolic function.

In most studies, the number of controller nodes in a genomic-scale model is usually considerably less than the number of driven or controlled nodes [32,114,128]. This observation is valid for several biological networks, being one of the bases for the development of analytical methodologies that allow it (such as the minimal set dominate methodology).

However, control coefficients are a fundamental concept in metabolic control methodology [130,131]. Control coefficients measure the relative change in metabolic flux in steady-state models [120,131]. To date, two main types of control coefficients have been proposed: the concentration control coefficient, which measures the relative change in metabolite concentrations in response to a change, and the flux control coefficient, which measures the relative change in the flow of matter in a metabolic pathway according to an evaluated scenario [131]. However, later, we show some of the most used examples of control coefficients.

This information provides insight into the dynamics and regulation of control in biological systems, offering the researcher inferences about the functioning and dynamics of the network.

Due to the heterogeneity of chronic diseases, the development and optimization of control theory have offered (provided) a robust mathematical framework for understanding (comprehending) various biological systems. The control methods for identifying the minimum set of controller nodes can be divided mainly into two categories (Table 1) [33,132,133]. According to [132], one category is associated with non-symmetric networks, and the other focuses on non-directed (or symmetric) networks. Depending on the type of model (dynamic or steady state), methodologies such as the Minimal Dominant Set (MDS) or Feedback Vertex Set (FVS) can be applied [26]. It should be noted that all the methods start from the stoichiometric matrix (*S*) as a common element.

Ref. [137] states that, regardless of the selected methodology, system controllability can provide results associated with the identification of chronic disease nodes and drug targets. In addition, ref. [33] highlights the utility of these methodologies in identifying viral proteins. At present, there are few approaches to investigate the controllability of systems and methodologies such as MDS that require high costs and, in some cases, may underestimate the structural capacity of the models [132].

Regarding cancer, ref. [138] explained that driver nodes can help identify driver genes but that it cannot be applied directly to the identification driver genes in a personalized manner for individual patients, creating a gap in personalized medicine. However, the classification and identification of driver nodes can be directly applied for the detection of driver metabolites in human liver models [139] and signaling pathways [140], as well as the identification of critical regulatory genes in cancer signaling networks [102]. Similarly, control methodologies have facilitated the analysis of biological models so that their structural controllability can be directly resolved; however, control methodologies need to be made more efficient and optimized. This adds to the challenge of assessing the validity of a network model due to its diverse objectives and applications.

In neurosciences, although there are few studies associated with the application, authors such as [141] have described the importance of computational methods that can infer the deregulation of key metabolites associated with neurons and microglia and that, in turn, can determine the potential basis for the development of diseases such as Alzheimer’s disease. Nevertheless, neuroscience and neuroengineering still face unique challenges due to the multiple interacting components that produce emergent behaviors [138]. In addition, to date, the approaches associated with control theory have been used to model high-level neurocognitive processes, that is, individual differences in creativity and intelligence, examining how the brain’s structural connectivity controls dynamic processes [142]. The controllability of the systems is a challenging issue, with the potential to describe the characteristics of omics data in neurodegeneration accurately and cellular influence on its progression [26,135,136]. Thus, applying these methods at the cellular level is necessary to recognize the dynamic characteristics of a biological system, such as neurons and astrocytes.

To make it easier to understand control methods, we provide an intuitive explanation of the two most used methods. Therefore, we summarize some key elements below.

#### 2.7.1. Control Coefficients and the Power Decay Law

Control coefficients are an important concept in metabolic control theory [128,143]. Control coefficients play an important role in the robustness analysis of metabolic networks [144]. According to [143], control coefficients help to quantify the robustness of metabolic pathways, since, in addition to determining controller nodes, they also allow for the evaluation of the sensitivity of the system to changes in enzymatic activities.

A general property of biological systems is the biostability capability, which assumes that bistable switches filter out slight and transient changes in the input signal that may even be typical in a homeostatic state [114,145]. This is important because this information, along with control theory, explores positive feedback architectures to generate uptake and repression rates.

Regardless of the model, like all in silico methodologies, the control structure must be taken carefully and evaluated according to the robustness criteria that allow it be compared with biological reality [146]. Among the parameters are (1) flow control coefficients, (2) elasticity coefficient (associated with the relative change in the activity of an enzyme), (3) concentration control coefficients, and (4) power decay law [29,147,148]. Since the elasticity and concentration coefficients are mostly associated with dynamic models, we focus more on the flow control coefficient and the power decay law in steady-state models.

The flow control coefficient is the degree of control pattern (*a*) of the activity that an enzyme (*i*) exerts on the flow in a pathway (*J*).
(4)$$C_{ai}^{J}=\frac{\%\;change\;in\;flux}{\%\;change\;in\;activity\;of\;enzyme\;i}\;$$

According to the authors of [145], the control coefficient is calculated from the slope of the tangent at the reference point. At the experimental level, it is calculated from the titration of two enzymes until a change in their activity is observed. Reactions that favor flow at the metabolic level tend to have a positive flow control coefficient value, while those that are not favored or are part of leak reactions have negative values [145]. The sum of all the values calculated for each reaction has a value of 1:(5)$${\sum C}_{ai}^{J}=1\;$$

However, one of the most studied parameters is the power decay law. In [29], the power decay law is considered according to the fraction of coupled reactions to quantify the predictive power of control theory in metabolic models at the genomic scale. This is because the systems’ complexity and metabolism depend on the degree of coupling, especially the reactions associated with central metabolism. However, the complexity of the controllability of systems is mainly independent of the number of couplings in networks that have few controlling reactions [26].

As an example, ref. [29] used the power decay law for 23 metabolic models (eukaryotic and prokaryotic), showing that the fraction of reactions decays logarithmically. The above indicates that reactions that do not affect other reactions (non-controlling) can be eliminated without directly affecting the optimization. As the authors show, the decay law allows the logarithmic curve to stabilize, indicating the number of minimum reactions for cell maintenance, making it a robust method.

The parameter of power decay law is associated with the activity pattern sampling methodology, meaning that the flow coupling analysis is not significantly altered by eliminating reactions. On the contrary, traditional methods such as FBA, sensitive to missing reactions [149], generate significant alterations in the metabolic phenotype, and, thus, they distance the results from biological reality. Nevertheless, it must be remembered that together these tests give analytical robustness, demonstrating the power of these types of methodologies to make predictions from incomplete network reconstructions [26,29].

#### 2.7.2. Feedback Vertex Set (FVS), a Dynamic Approach

Control theory methods associated with systems biology are based on the principle of intervention and the control of dynamic networks. Authors such as [135,136] note that they use a system of nonlinear functions where the ordinary differential equations (ODEs) give a directed graph.
(6)$${\dot{z}}_{k}=F_{k}\left(t,z_{k},\;z_{I_{k}}\right)\;$$
where *F* defines a nonlinear function, k=1,…,N, and $I_{k}\subseteq \{1,\ldots ,N\}$ are given subsets. However, the function does not necessarily have to be dependent on time. In fact, due to the nature of this ODE system, the solution of $\frac{\delta }{\delta t}z_{k}$ enters into a Euclidean ball around the origin of a sufficiently large fixed C, where solutions exist globally in forward time [150,151]. Nevertheless, an assumption is that the global model has a decay condition; this means that function F has the condition that $\delta_{1}F\left(z_{k},z_{I_{k}}\right)<0$, and the previous condition is necessary to find a set of determining nodes that will be given when the difference in the two solutions of the nonautonomous ODE systems tend to zero and when t tends to infinity [135].

The previous means that, given two solutions of the ODE system, $z_{k}\left(t\right)$ and $z_{k}^{\prime }\left(t\right)$, the determining nodes will be defined when the difference tends to zero and when the time tends to infinity; this means that
(7)$$z_{k}\left(t\right)-z_{k}^{\prime }\left(t\right)\;\underset{\;t\;\to \;\infty \;}{\to }\;0\;$$

From the above, it is possible to find a convergence over the system to identify determining nodes that are sufficient to determine large-time dynamics. In addition, the dynamic approach enables the detection of steady-state, periodic, and quasi-periodic solutions.

#### 2.7.3. Minimal Dominant Set (MDS) and Probabilistic Blocking of Metabolic Fluxes Approach

The previous methods depend on a solution of nonlinear characteristics, such as the calculation of the coefficients of elasticity associated with kinetic models of metabolic pathways [135,136]. However, due to the lack of information about dynamic models over time, an approach associated with the types of flow couplings between reactions in steady-state models was recently developed [32]. Ref. [134] proposed this method to analyze and identify metabolic switches to modulate complex diseases. The method called Minimal Dominant Set (MDS) assumes the controllability of a system from a small set of reactions (or nodes) that must be controlled indirectly to control the activity of all reactions in the system [26,29,134].

According to authors such as [32], analyses based on stoichiometric and thermodynamic restrictions could limit the response range of a metabolic model. The above, according to [70], gives rise to the formation of the solution space as described in the following equation:(8)$$F\left\{v\;\in \;R^{n}|\;S\;v=0,\;lb\leq v\leq ub,\;\exists i\;\epsilon \;E:v_{i}\neq 0\;\right\}$$
where *S* is the stoichiometric matrix of size m∗n, with *m* being the metabolites and *n* being the reactions; *ub* and *lb* are the upper and lower limits; *v* is the flux vector; and *E* is the set of exchange reactions. Also, *Rev* = {1,…, *n*}∖*Irr* is the set of reversible reactions, and *Irr* ⊆ {1,…, *n* } is the set of irreversible reactions.

In control theory, optimization methods allow for the calculation of coupling relationships, unlike FBA, which determines only the metabolic phenotype of a biological model according to its objective function [21,59,152]. One of the most used methods is flow coupling analysis (FCA), which allows for the calculation of the possible flow distributions and the coupling relationships between reactions [153]. Recently, authors such as [70] proposed the fast flow coupling calculator (F2C2) to determine the coupling between reactions. These methods are based on linear programming problems, where feasibility rules eliminate trivially uncoupled reactions [70,154,155]. However, authors such as [154] also use optimization methods such as the Mixed-Integer Linear Program (MILP) to determine the type of coupling between reactions.

Regarding the coupling profiles of a genomic-scale model obtained in the vector ϕ ∈ ℝ*^n^*, these profiles are grouped in pairs considering the average Euclidean distances between each flow coupling profile, as described in the following equation:(9)$$\sum_{i=1}^{n}\frac{\left(x_{i}-\mu_{x})(y_{i}-\mu_{y}\right)}{\sqrt{varx}\sqrt{vary}}=\frac{Cov\;(x,y)}{\sqrt{varx}\sqrt{vary}}$$

The above describes the possible relationship between the couplings, which allows for the application of grouping algorithms, such as k-means and k-medoids; that is, to build the relationship between the reactions, an approximation via K-means or K-medoids is used. The authors of [29] also highlight the utility of performing metabolic network randomizations to analyze whether the coupling profiles reflect functionally relevant features of metabolism using z-scores:(10)$$z_{i}=\frac{\left(x_{i}-{\underline{y}}_{i}\right)}{\sigma_{i}}$$
where $x_{i}$ is the relative frequency of coupling in *S*, and ${\underline{y}}_{i}$ is the average frequency of type *i* coupling.

These randomizations also allow for the random sampling of the flow distributions of a model in the steady state, eliminating the bias of the use of an objective function and enabling the observation of all the possible scenarios and the importance of each reaction in the model, which allows the patterns of activity of each reaction to be obtained.

The above can be synthesized in a control graph such as the following [29]:$\sigma_{i}=\sigma_{j}=1$ and $L\left(i,j\right)\in \left\{full,\;partial,directional\right\}$, or$\sigma_{i}=0,\sigma_{j}=1$ and $L\left(i,j\right)=anti$, or$\sigma_{i}=1,\sigma_{j}=0$ and $L\left(i,j\right)\in \left\{inhibiive\right\}$, or$\sigma_{i}=\sigma_{j}=0$ and $L\left(i,j\right)\in \left\{full,\;partial\right\}$, or$\sigma_{i}=\sigma_{j}=0$ and $L\left(i,j\right)=directional$,and $M_{i,j}=0$ otherwise.

Here, the state or pattern of reaction activity *σ* is given by the generated flux vector *v*.

However, authors such as [32] have determined that MDS uses a discrete definition to identify the coupling between reactions when performing randomizations for grouping activity patterns. Therefore, these authors propose the use of the angles formed between the product of the transposed matrix *K* (or Kernel) and *S*, obtaining in terms of squared cosine the Pearson correlation between reaction flows *ij* or correlation coefficient $\theta_{ij}^{}$, which allows the use of these measures of a linear relationship between variables to define the relationship between reactions. This methodology was proposed by [156], coupled with a probabilistic control model. Therefore, the value of the correlation coefficient is comprised of values between 0 and 1, where 1 is a high correlation indicating a strong connection between the two reactions. In contrast, a value close to zero indicates that there is no relationship, and, therefore, the connection between reactions is low or nil. However, the authors define the probability of failure between two reactions as $\rho_{ij}^{}=1-abs(\theta_{ij})$ and integrate this into a probabilistic minimum dominant model (PMSD) using the probabilistic blocking of metabolic fluxes approach [32]. The above can be described using the following equation:(11)$$(1-\prod_{i\;\epsilon \;S}^{}\rho_{ij}^{}\;)\;\geq \;\Theta \;ó\;\sum_{i\epsilon \;S}^{}-ln\;(\rho_{ij}^{})\;\geq -ln\;(1-\Theta )$$

Here, $\rho_{ij}^{}$ defines the probability of failure between reactions *i* and *j.* According to the above, the solution to a linear problem is established where the minimization of
(12)$$\sum_{i\in V}x_{i};con\;x_{i}\in \left\{0,1\right\}\mathrm{subject}\mathrm{to},$$



$$x_{j}\geq 1,\;\forall j\in V\;such\;that\;degree\;\left(j\right)=0,$$


(13)
$$-\ln\left(1-\Theta \right)x_{j}+\sum_{i,j\in E}\left(-\ln\left(\rho_{ij}\right)x_{i}\right)\geq -\ln\left(1-\Theta \right),\forall \;i\in V\;such\;that\;degree\left(j\right)=0$$



The authors describe the above in such a way that this methodology seeks to find the shortest path between the reactions *(i, j*), where they have a greater probability of coupling if the linear relationship between them is close to 1 [32]. It is important to highlight that this methodology of probabilistic blocking of metabolic flows is similar to the traditional MDS approach, since both use an LP approach to optimize and find the possible types of coupling. Both methodologies allow for defining a distance measure between nodes to find a relationship between them. It should be noted that these methodologies have been widely used for the construction of control models in cancer to identify controller nodes and possible biomarkers [19,32,103,129,157,158].

## 3. Machine Learning (ML) and GEM Approaches to Determine Metabolic Markers

GEMs provide a quantitative tool to establish the relationship between genotype and phenotype by contextualizing various types of Big Data, such as genomic, metabolomic, and transcriptomic data. The simulation of metabolic fluxes, crucial in flux rate prediction, is effectively executed through GEMs [86,87]. GEM reconstruction and analysis have proven fundamental for a deeper understanding of metabolism in diverse organisms, including bacteria, archaea, and eukaryotes [23,86]. The versatility of GEMs is reflected in their application to a wide range of species. Furthermore, integrating omics data into these models is essential for standard GEM analysis, improving flux predictions and allowing for a more accurate interpretation of multi-omics data.

In the 1950s, Alan Turing laid the foundation for machine learning in artificial intelligence with his article “Computing Machinery and Intelligence” [159], despite the computational limitations at the time. With the increase in datasets and computational power, significant advances have been made in addressing challenges such as managing large amounts of data [160]. In this context, the machine learning (ML) field has seen notable progress in recent years. ML involves using algorithms to handle large volumes of data and answer research questions, identifying patterns, trends, or anomalies [161] using classification or regression algorithms. Various strategies are used depending on the research objective, such as unsupervised learning, which discovers patterns in unlabeled data, and supervised learning, which uses labeled examples to make predictions for new data [162,163].

ML has been used in biology to generate models that extract information from complex and voluminous datasets. In the context of genome-scale metabolic models (GEMs), various strategies have been explored in applying ML to improve the understanding of patterns and characteristics in complex biological data. For a few years now, approaches to the use of ML with GEMs have been made under three strategies: (1) fluxomics, (2) multimodal analysis, and (3) the generation of models based on constraints together with fluxomic data [164,165].

Fluxomics, a term coined in the last two decades, refers to studying metabolic fluxes in cells, tissues, or organisms [166,167]. The intersection between fluxomics, genome-scale metabolic models (GEMs), and machine learning (ML) is a constantly growing field, providing a deeper understanding of the interactions between metabolism and genetics and facilitating prediction, analysis, and metabolic pathway engineering [168].

The application of ML in conjunction with fluxomics and GEMs has significantly improved predictive performance and data coverage. Fluxomics is a rapidly growing field applied in various research areas, including biotechnology, pharmacology, and metabolic engineering. A notable example of the application of fluxomics is in the study of [169], where a computational procedure was proposed to translate metabolite profiles (metabolome) into metabolic fluxes (fluxome). This method uses computational models integrated with linear optimization, dynamic and continuation analyses, and metabolic control. Tests were performed on metabolite profiles obtained from ex vivo mouse Langendorff heart preparations perfused with glucose to validate the procedure.

Another relevant example in fluxomics is the study in [170], where FASIMU (version 2.3.4), flexible software for flux equilibrium calculation series in large metabolic networks, was developed. This software was used to analyze the metabolic network of Escherichia coli, and the results demonstrated that FASIMU is effective in simulating the metabolic behavior of large-scale metabolic networks.

Furthermore, fluxomics has been instrumental in drug discovery, as illustrated by the study in [171]. These researchers used fluxomics to target bacterial metabolic pathways distinct from human ones. The approach consisted of identifying the metabolic pathways essential for the survival and growth of bacteria using fluxomics and then developing therapeutic strategies based on the selective inhibition of these pathways with antibiotics.

When analyzing multiple omics data, such as transcriptomic, proteomic, or metabolomic data, processing is performed to synchronize and obtain fluxomic information from a model restricted to a specific condition [162]. Despite its benefits, the use of ML presents challenges. The reproducibility of the models is essential, requiring the creation of well-defined reference points and references for their use in the scientific community [172]. Building ML models requires the careful selection of data and the features or patterns of interest, model construction, validation, and the consideration of limitations and biases for an accurate interpretation of results [163].

Although ML has been applied in various areas, its integration with genomic-scale metabolic models to identify biomarkers in neurodegenerative diseases has yet to be carried out. This integration could accelerate research, save time and resources, and avoid ethical issues when operating in silico, opening opportunities for system-based therapeutic interventions to address simulated neurodegenerative mechanisms.

## 4. Overview of Systems Biology Applications with Control Theory in Chronic Diseases

Although most studies have focused on employing systems biology in unicellular organisms, in recent decades, an increasing number of works have implemented this methodology in human cells to understand their functions [4,173]. Moreover, the evolution of this approach has allowed for the implementation of other methods, such as control theory, in the construction of GEMs and networks in various research fields, including cancer and neurosciences (Table 2).

In the field of cancer research, it is noteworthy that control theory has been implemented more than in neuroscience or other areas of research, and it has even been applied for drug target searching. Several studies have corroborated that this is a powerful tool to identify driver reactions that control a system [29]. For example, in 2013, Asgari and collaborators performed one of the first works employing control theory; they analyzed 15 GEMs from normal and cancer tissues to identify drug targets. In this work, a centrality analysis was used to evaluate whether known drug targets are highly connected nodes in the network, and they employed topological parameters to find driver nodes; both approaches were proposed considering the MDS concept. The centrality analysis did not show drug targets as driver nodes controlling the systems. However, by applying topological parameters, such as motifs and clusters, the authors found that drug targets belong to a specific cluster of an enzyme-centric network. Based on these results, the authors suggested that more complex metrics could be relevant for assessing controllability [101]. Another study in 2016 analyzed four cancer networks by applying the flux coupling method. The authors found that, by clustering the flux coupling profile of cancer networks, healthy tissue can be discriminated from cancer samples. Moreover, they found that the identified driver reactions are associated with genes that cause cancer. Hence, Basler and collaborators highlighted the relevance and usefulness of this method, since previous works only assessed the absence or presence of reactions in cancer versus healthy samples without identifying the reactions driving disease development [29].

However, another analysis was conducted using the same GEMs used by Basler and cols. Ref. [29] in order to observe the metabolic implication of the transition from a healthy state to a pathological state using the probabilistic minimum dominating set (PMDS). Interestingly, differences between cancer and heathy networks were not observed using topological analyses; however, a lower rate of driver nodes was detected in cancer than in healthy cells using PMDS, indicating that it is easier to control metabolic networks. In this case, the authors used two approaches, control theory and flux correlation. These results underscore the ability of control theory to detect differences between healthy and disease states in comparison with other methods [32]. A noteworthy finding is that affected cells had lower driver nodes; the same observation was made in a study that assessed the interactome of cells infected with a HIV virus [33], which could indicate that pathological states are triggered by reducing the complexity to control a system.

Furthermore, it is important to highlight that, in a study about breast cancer, the power law degree was employed to identify high-degree nodes in GEMs; high-degree nodes are nodes that dominate and play vital roles in the network. In this research, after identifying the hub nodes, the authors applied additional statistical methods; these methods were correlation and principal component analyses, which led to the identification of five top drug targets [174].

There are many challenges in characterizing and stratifying different types of tumors, along with identifying effective cancer treatments. In this context, in 2018, Bidkhori and colleagues [176] performed two research works focusing on hepatocellular carcinoma. In the first work, a GEM was constructed considering the transcriptomic data of cancerous liver samples and non-affected tissue. Subsequently, the authors applied a controllability approach to identify exclusive controlling genes in cancer tissue and non-affected tissue. Moreover, discrimination of different subtypes of hepatocellular carcinoma was carried out by identifying controller genes in the metabolic networks, which might be crucial for identifying therapeutic targets. Two aspects stand out in this study; the use of both objective-dependent and objective-independent methods, and the use of GEMs to generate a functional gene–gene network for identifying candidate targets using the control theory approach [176]. In the second work focused on hepatocellular carcinoma, the authors aimed to identify and prioritize anticancer targets using the same strategy of combining two methods. They initially constructed GEMs using expression data to obtain personalized metabolites and reaction networks; then, they applied a controllability analysis to determine driver nodes. In the first step, they identified 374 antimetabolites that were reduced to 142 controlling metabolites in the second analysis, allowing for the final selection of 74 anticancer controlling metabolites. These results suggest that the combination of objective-dependent and -independent methods allows for the identification of suitable therapeutic targets. This prioritization was important, because the authors found that a large number of nodes with high centrality can either result in the lethality of both healthy and tumoral tissues or not result in lethality, indicating that a great amount of data can sometimes generate undesirable information that can lead to misinterpretation [176,177]. Therefore, it is relevant to consider the valuable results derived from control theory in comparison with other methods, as this might provide more precise data.

As previously shown, systems biology is now very useful in identifying and distinguishing the metabolic characteristics in normal and affected tissues or in patients with a disease and healthy subjects, which can be relevant to find biomarkers or effective drug targets. In addition, control theory can also be applied to investigate the metabolism of a particular organ or system. For instance, Liu and Pan (2014) [139] used the controllability analysis in a GEM of liver that was built based on knowledge from the literature and transcriptomic, proteomic, metabolomic, and phenotypic data. In this work, they classified 36 critical driver metabolites and 27 essential metabolites; the driver metabolites were all associated with transport reactions in the extracellular compartment.

Control theory has been used sparingly in studies of systems biology in neuroscience; however, it is important to highlight the need to implement this approach, especially for studying neurodegenerative disorders like Alzheimer’s disease, one of the most prevalent diseases in older people, generating a significant disease burden [178]. This implementation could be highly valuable considering that there are many challenges in finding effective treatments and biomarkers for neurodegenerative diseases, and huge efforts have been made to obtain biological insights that allow for early diagnosis in order to prevent or delay disease progression; however, the mechanisms are still not well known [179,180]. It is relevant to mention that systems biology and GEMs have been implemented in different works in neurosciences [181]. Classical methods like FVA have been very useful in integrating data from transcriptomic analysis to identify reactions altered in three psychiatric diseases (schizophrenia, bipolar disorder, and major depressive disorder) compared to controls, suggesting potential candidate biomarkers [54]. Nevertheless, the classical methods have some limitations and criticisms, which include the lack of dynamic and regulatory aspects in the models [106].

Control theory has been used in various works in neuroscience, primarily focusing on brain connectivity; these works have used this approach to study metabolic brain networks by means of positron emission tomography in humans, for example, in people with subjective cognitive decline [182] and in schizophrenia patients using functional magnetic resonance imaging [121]. However, only one study has implemented this approach in genome-scale models; that work was carried out considering the integration of transcriptomic and proteomic data from an in vitro model of astrocytes [34]. Similar to other studies, this work used a combination of methods, including FBA, FVA, and MDS analyses, which reveled metabolic changes induced by a palmitic acid stimulus, identifying perturbation of the folate cycle, fatty acid β-oxidation, and 25 other metabolic switches [34]. The identification of these metabolic switches (driver nodes) could be very useful for identifying drug targets and potential therapies for diseases triggered by inflammatory factors, such as neurodegenerative diseases. Moreover, this study introduced a novel application of control theory to integrate data from in vitro models, offering an excellent opportunity to understand cellular function and responses to different challenges. In addition, it might be the basis for understanding interactions among brain cells [181].

Although control theory has not been widely used, these examples offer a new perspective about its further applications. Moreover, it is worth noting that this method might be helpful in scenarios where sample size limitations could impact the statistical analysis [26], and it may even be valuable in developing therapeutic strategies tailored to specific patients [122].

## 5. Potential Applications in Neurodegenerative Disorders: A Focus on Alzheimer’s Disease

As we can see above, control theory has mainly been applied in studies on pathologies like cancer, but in neurodegenerative diseases, it has not been explored. However, it is worth mentioning that many efforts have been made to analyze multi-omics data derived from samples of patients with Alzheimer’s disease. For instance, a metabolic network was obtained with a constraint-based model using differential gene expression data derived from the postmortem tissue of 1708 samples. In this study, the authors noted that the metabolic flux differs mainly between AD and mild cognitive impairment but not with control samples using the FVA analysis. Specifically, these differences were observed for three enzymes in the dorsolateral prefrontal cortex, the serine palmitoyltransferase (SPT), sphingomyelin synthase (SMS), and ceramide kinase (CERK) [183]. This finding could be a key element for identifying the characteristics that determine the progression of a phenotype to the development of a disease, with this being quite relevant for cases of mild cognitive impairment, which is a cognitive disturbance that leads to AD in about 15% of cases [184]. In this context, although control theory has not yet been explored, this example highlights the potential application of systems biology in the identification of mechanisms associated with the early stages of AD [185].

## 6. Future Directions

In summary, all the above illustrates the power of control theory for studying many aspects of human systems, such as exploring biological functions in cell-type-specific ways and examining metabolic perturbations under challenging conditions or pathologies. Furthermore, regarding human diseases, studies have shown that systems biology coupled with control theory is highly useful for the identification of biomarkers and drug targets and for classifying different disease states or tumor subtypes in patient samples (Figure 5A). Interestingly, there is also evidence of the ability to detect responses to treatments, which could be of great relevance for personalized medicine. Moreover, this approach opens new possibilities to analyze and interpret the vast amount of data derived from high-throughput techniques, which has presented a great challenge in most recent years.

A relevant aspect evidenced in studies applying control theory is that the most reliable and significant findings can be obtained by combining different methods, which involve the classical tools for GEM reconstructions followed by the implementation of control theory, topological properties, and statistical analyses (such as correlation and principal component analyses), among others (Figure 5B).

Finally, while the implementation of this method has been mainly performed in the cancer research field, preliminary studies in neurosciences suggest that implementing this tool will be useful in identifying biological and metabolic markers in the early and prodromal stages of pathologies such as neurodegenerative diseases, where an early diagnosis is required because their symptoms are highly nonspecified. Therefore, we encourage further studies to leverage this method to enhance our understanding of the development of neurodegenerative diseases.

## 7. Materials and Methods

We carried out a review that investigates control theory, systems biology, and their applications, especially in neurodegeneration. Articles eligible for inclusion were articles published in indexed journals that characterized the methodology and application in chronic or complex diseases.

Therefore, we designed a highly sensitive search strategy that combines free text, logical operators, and the following combinations of generic groups: systems biology, control theory, and applications of control theory with an emphasis on biological systems; neurodegeneration and systems biology; and neurodegeneration, systems biology, and control theory. We then systematically searched for articles published on Google Scholar. Additional searches were performed on servers such as Scopus, ScienceDirect, and PubMed. The selected articles refer to works presented in peer-reviewed journals.

For greater sensitivity, we used the following search terms:Control theory OR Systems biology;Metabolic models on a genomic scale OR GEMS AND Control theory;Minimal dominant set (MDS) approach OR MDS OR Control theory AND Systems biology;Probabilistic blocking of metabolic fluxes approach AND Systems biology AND Control theory;Metabolic models on a genomic scale AND Control theory AND applications OR Neurodegeneration;GEMs OR genomic-scale metabolic models OR systems biology AND control theory AND applications OR neurodegeneration OR chronic diseases OR complex systems OR applications in neurodegeneration AND date limit 2000/01/01–2023/08/01.

No exclusions were made for disease severity or reported outcomes. An additional search was performed in the application research rabbit app in order to identify other possible related works.

## Figures and Tables

**Figure 1 ijms-25-00365-f001:**
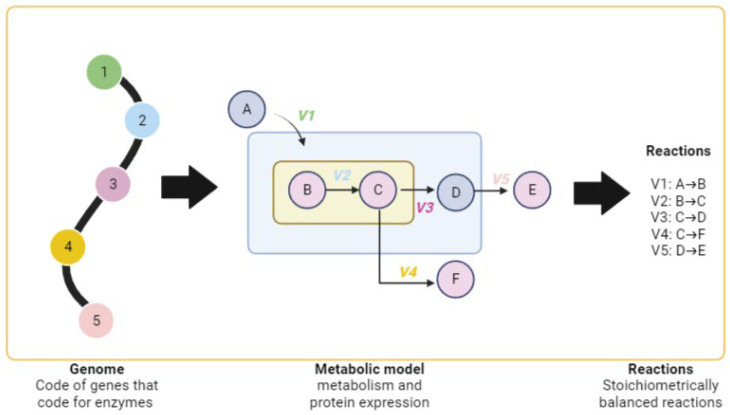
The systems biology approach to genomic-scale model reconstruction. The letters A–F represent the metabolists included in the system.

**Figure 2 ijms-25-00365-f002:**
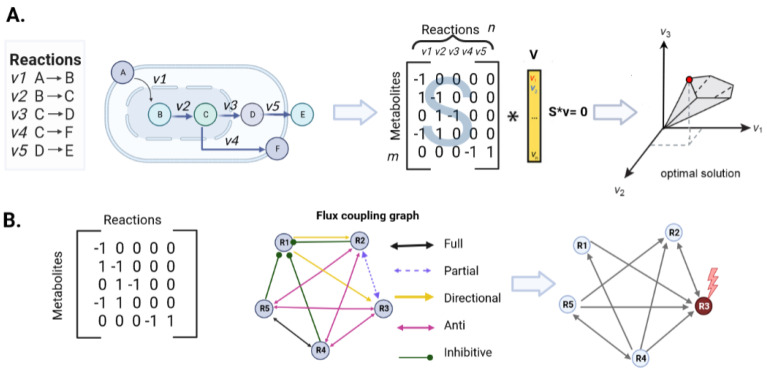
Flow balance analysis (FBA) scheme and representation of the flow coupling graph. (**A**) The metabolic network consists of a list of stoichiometrically balanced biochemical reactions (R1, R2, R3, Rn). This type of reconstruction can be represented mathematically by a stoichiometric matrix (*S*) of size *m* × *n* composed of its reactions and metabolites. The steady-state flow distribution is defined by the equation *S*v* = 0, where *v* is a flow vector. According to the objective function of interest, maximization or minimization is performed; optimization allows for the finding of the flux distribution that enables the optimal solution for this objective function while observing the restrictions given the principle of mass balance and the limits of reaction [21]. (**B**) The application of control theory starts from the use of matrix *S*. The vertices represent reactions, and the labeled edges represent the five coupling relationships (represented by different colors; see legend). The steady-state principle implies that some reactions operate in a concentrated manner, leading to reaction coupling relationships. Reaction *i* is directionally coupled with *j* if *σj v* = 0 implies *σi v* = 0. Partial coupling is a particular case of directional and full coupling: two reactions, *i* and *j*, are partially coupled if they have the same state. If one of the two reactions is inactive, then a steady-state flux is only possible if the other reaction has a non-zero flux, which would define two anti-coupled reactions. Finally, reaction i couples inhibition with reaction j if the maximum flux of reaction i implies that j is inactive. The above can be summarized as follows: (1) the directional and total (or active) flow of R1 leads to the activation of R2 and R3 and the inactivation of R4 and R5; (2) the inactive flow of R1 leads to the deactivation of R2 and the activation of R4 and R5; (3) the inactive flow of R1 leads to the activation of R2 and the deactivation of R4. Taken and modified from [29].

**Figure 3 ijms-25-00365-f003:**
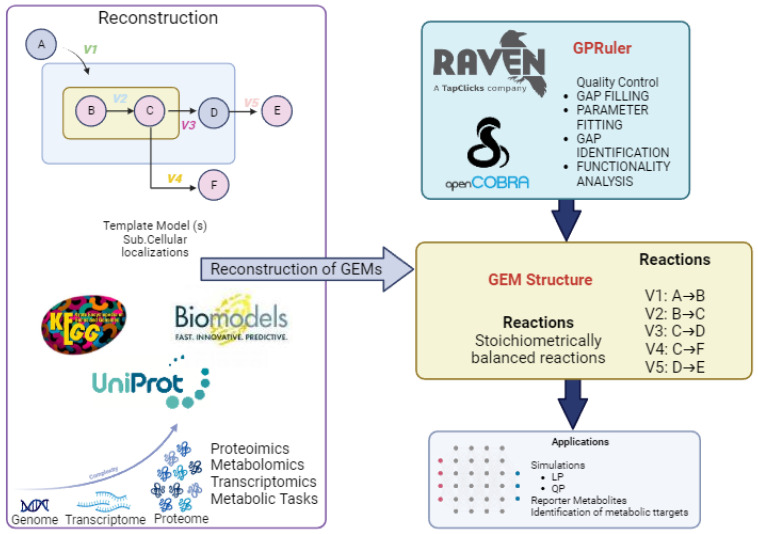
A simplified representation of the steps required in the construction of a GEM. This process involves the use of various databases, including but not limited to KEGG and UniProt, and it may involve a reference model.

**Figure 4 ijms-25-00365-f004:**
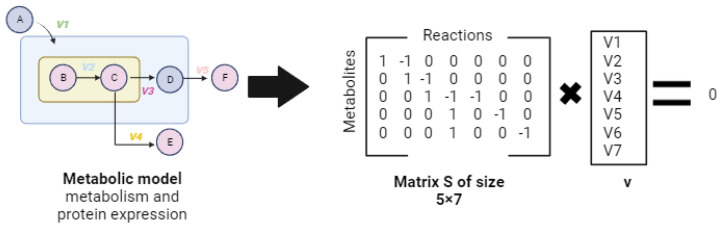
Matrix *S* of size 5 × 7 illustrates the proportionality of the chemical species at a given time, multiplied by the flux vector. This flow vector is the set of paths participating in the objective solution. The result is the calculation of a vector of zeros, which allows us to assume an equilibrium state of the model where there is flow in the system. However, there are no changes in the concentration of the metabolites.

**Figure 5 ijms-25-00365-f005:**
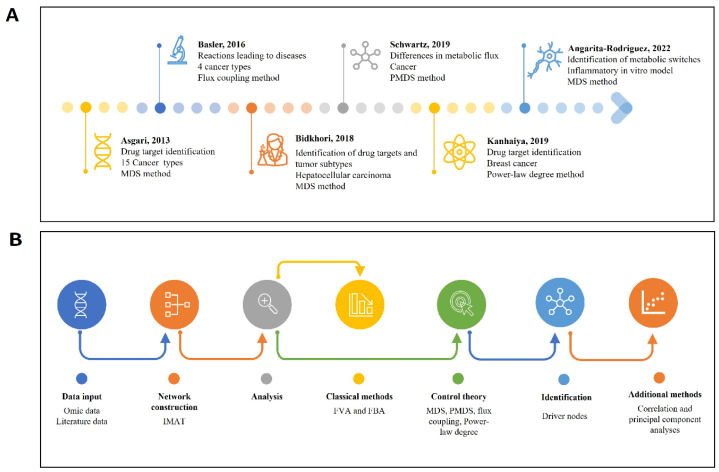
Applications of systems biology with control theory. (**A**) Timeline of studies using control theory to identify drug targets or metabolic perturbations mainly in cancer [29,32,34,101,174,176]. (**B**) Diagram showing the most common workflow evidenced in studies employing control theory for analyzing metabolic network models.

**Table 1 ijms-25-00365-t001:** Description of the theory methods used to identify controlling reactions in systems biology.

Methods	Authors	Network Styles	Dynamics	Input	Principle	Optimization
MDS	[29,134]	Undirected networks	Nonlinear	Adjacency matrix/ stoichiometric matrix	Optimization based on KNN and K-means using Euclidean distances and Bootstrap.	Linear discrete
Probabilistic controllability approach	[32]	Undirected networks	Nonlinear	Adjacency matrix/ stoichiometric matrix	Optimization based on the shortest path between nodes based on an index built using the correlation between reactions.	Linear continuous
DFVS	[135,136]	Directed networks	Nonlinear	Adjacency matrix/ stoichiometric matrix	Optimization based on nonlinear functions in a system of ODE that define the structure of the graph.	Nonlinear continuous/ discrete

**Table 2 ijms-25-00365-t002:** Applications of systems biology with control theory.

Author	Outcome	Sample Type	Omics Data Type	Main Outcome or Application	Method
[34]	Human astrocytes stimulated with palmitic acid	Cultured cells	Transcriptomic and proteomic	Identification of metabolic switches	FBA, FVA, MDS
[174]	Breast cancer	Cancer and normal samples	Based on previous GEMs [175] Transcriptomic and proteomic	Identification of drug targets	Topological properties and power-law degree
[32]	Healthy and cancer states	Tissues	Based on previous GEMs (Basler et al. [29] and Gatto et al. [175])	Differences in metabolic flux	PMDS
[176]	Hepatocellular carcinoma patients	Cancerous liver samples and non-affected tissue	Transcriptomic	Detection of tumor subtypes	FBA, minimum driver node sets—MDS
[29]	Breast, lung, renal, and urothelial cancers	4 cancer and healthy sample networks	Based on previous GEMs [175] Transcriptomic and proteomic	Reactions leading to cancer	Flux coupling
[101]	15 cancer types	Normal and corresponding cancer cells	Transcriptomic	Identification of drug targets as driver nodes	MDS

## Data Availability

No new data were created or analyzed in this study. Data sharing is not applicable to this article.

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
