# Peer review of "Control Theory and Systems Biology: Potential Applications in Neurodegeneration and Search for Therapeutic Targets"

_ijms, 2023, doi:10.3390/ijms25010365_

Round 1

Reviewer 1 Report

Comments and Suggestions for Authors

The review explores common control methods used with genomic-scale metabolic models in steady state, focusing on multi-omics. It also discusses their applications in human chronic diseases and suggests potential extrapolation to processes of cognitive deterioration. Τhis review is not only useful for scientist deal with systems biology with control theory but also the people out of this area. So, I recommended to accept this review after following changes. Some improvement and additions could be included:

GEMs have been be reconstructed using automatic and semi-automated tools? How can be predicted the growth rate of the organisms? Though simulating the metabolic fluxes in the system or not?

The determination of effective exchange reaction constraints could be discussed, mainly when building a GEM as the first challenge is collecting enough extracellular metabolomic data.

The authors could stress machine learning approaches (supervised, unsupervised) used to better analyses outputs and extract meaning from complex model predictions, ML classifiers for metabolic markers determination.

It is not clear how applying topological parameters such as motifs and clusters, identification of drug targets to a specific cluster of an enzyme-centric network can be performed.

A classification of models for interactions among gene, protein and drug molecules into static network and dynamic modeling could be discussed. Some examples of constructing a static network and limitations could be included. Also, drug administration with control theory could be stressed.

Specific targets for future studies and directions could be included.

Comments on the Quality of English Language

Minor editing of English language required

Author Response

Consulte el archivo adjunto

Reviewer 2 Report

Comments and Suggestions for Authors

This paper is too abstract to be convincing.  While the authors describe the possible mathematical tools available to describe the control loop between disease progression and feedback to modulate the progression, I find it difficult to see how it would work in Alzheimer's Disease (say).

For AD for example, chronic inflammation is a sign that the wound is not healed. That is the wound healing process (feedback) fails and the tissue does not return to homeostasis. 

Usually, it is the breakdown of the feedback loop which allows the pathology to increase uncontrollably, leading ultimately to neurodegeneration.

The paper can do justice by providing a concrete example of a neurodegenerative disease. Otherwise it is all plausible but unconvincing.
